# Competence in Daily Activities and Mental Well-Being among Technical Intern Trainees in Japan: A Cross-Sectional Study

**DOI:** 10.3390/ijerph19063189

**Published:** 2022-03-08

**Authors:** Kuniko Arita, Akira Shibanuma, Rogie Royce Carandang, Masamine Jimba

**Affiliations:** 1Department of Community and Global Health, Graduate School of Medicine, The University of Tokyo, Tokyo 113-0033, Japan; kuniarita@m.u-tokyo.ac.jp (K.A.); rcarandang@m.u-tokyo.ac.jp (R.R.C.); mjimba@m.u-tokyo.ac.jp (M.J.); 2Department of Occupational Therapy, The Graduate School of Human Health Science, Tokyo Metropolitan University, Tokyo 116-8551, Japan

**Keywords:** Japan, mental health, migrant workers, occupational therapy

## Abstract

Migrant workers are at a greater risk of having low mental well-being compared to their local counterparts. The Japanese government accepts migrants through its Technical Intern Training programs, but the factors associated with their mental well-being remain unclear. This study aimed to (1) assess trainees’ competence and importance in daily activities and (2) examine the association between their competence and mental well-being. We conducted a cross-sectional study using self-administered questionnaires. We recruited trainees through their supervising organizations and social media. We used the Occupational Self-Assessment tool to measure competence and importance in daily activities and the World Health Organization-5 Well-being Index to measure mental well-being. Hierarchical regression analysis was used to examine the association between competence and mental well-being. Among 383 trainees, 30.6% felt difficulty expressing themselves, and 27.4% felt difficulty accomplishing goals. Almost 50% valued self-care, working towards their goals, and managing their finances. Higher competence scores were associated with higher mental well-being scores (B = 0.76; 95% CI = 0.52, 1.00). Competence may be a key to having higher mental well-being among migrant trainees in Japan. A supportive and enabling environment, as well as mental health promotion at the community level, may improve trainees’ competence in daily activities.

## 1. Introduction

Migrant workers experience a lower level of mental well-being than non-migrant workers in host countries [1,2]. For example, they tend to have a higher risk of depression [3] and suicide than non-migrant workers [4]. This is because migrant workers are exposed to migration-related factors, such as language barriers [5], new cultural norms [6,7], loss of social and familial support, and discrimination [7]. Moreover, they are exposed to work-related risk factors, such as greater job insecurity [8] and greater work-related hazards [9].

To overcome these migration-related risk factors, a level of occupational competence might be useful. Occupational competence is defined as the perceived ability to perform daily activities [10]. In the field of occupational therapy, “occupations” are defined as purposeful and meaningful activities that are typically categorized as self-care, work, play, leisure, and rest [11]. Occupational competence is not limited to work-related skills, but rather refers to all activities of daily living. For example, it includes taking care of oneself, maintaining a routine, helping significant others, and enjoying oneself and relaxing [10]. A higher level of occupational competence is associated with higher levels of mental well-being among non-migrant populations, such as mothers and university students [12,13]. In contrast, injured workers with low occupational competence experience more stress, anxiety, and depression [14,15]. For migrants, occupational competence may also impact their mental well-being when they experience changes in physical, economic, political, social, and cultural environments [16]. However, occupational competence and its role in mental well-being have not been examined in-depth among migrant workers.

Under the Technical Intern Training Program, a short-term labor rotation system in Japan that started in 1993, technical intern trainees are accepted into Japan as migrant workers. The Technical Intern Training Program is designed for foreigners to enter into an employment relationship, for up to five years, with a company, private business owner, or other training provider in Japan. Trainees aim to acquire, master, or become proficient in skills difficult to acquire in their home country [17]. Major job areas include agriculture, fishing, construction, food manufacturing, textiles and clothing, and machinery and metal processing. In recent years, trainees such as hotel clerks and those in the field of nursing care have been increasingly accepted. In 2019, of a total of 1.7 million migrant workers in Japan, 410,972 were registered trainees. Of them, 50% accounted were Vietnamese trainees, 23% were Chinese, and 9% were Filipino [18].

Specifically, migrant trainees may experience greater migration- and work-related risk factors than general migrant workers in Japan. For example, they might be exposed to the loss of social connections due to family separation [19], while other working visa holders can bring their family members to Japan [20]. Moreover, trainees cannot choose where to live, while other migrant workers or international students can decide to live close together, which often forms a community that serves as the foundation for social support. The program in the agricultural field limits the number of trainees to one or two per farm [21]. They also experience legal restrictions, such as a limited length of stay or not being allowed to change jobs [22].

Most previous studies have focused on risk factors; however, limited studies are available on migrant workers’ competence in terms of positively protecting their mental well-being. Only language proficiency in the host country has been identified as a key personal skill that positively affects mental well-being [23]. However, few studies have focused on migrant workers’ competence in daily activities in host countries and its association with mental well-being. Little is known about the daily activities that trainees perceive as important in a Japanese context. Therefore, this study aimed to analyze the levels of competence in daily activities and their importance. It further investigated the association between competence in daily activities and mental well-being among Chinese and Vietnamese technical intern trainees working in Japan.

## 2. Methods

### 2.1. Study Design and Setting

We conducted a cross-sectional study across all prefectures in Japan. Technical intern trainees reside in all 47 prefectures across Japan. In Japan, Aichi Prefecture had the highest percentage of trainees (39,600, 10.2% of the total). Even Akita Prefecture, with the fewest number of trainees, had 1355 registered trainees [18].

### 2.2. Participants

The eligible participants were Chinese and Vietnamese technical intern trainees working in Japan during the survey period (11 August 2020 to 30 September 2020). Participants needed to be native or fluent in Chinese or Vietnamese to respond to their recent feelings and competence in daily activities. We excluded those who did not possess technical intern trainee visas or who stayed in Japan for less than 14 days.

We calculated the sample size based on the following parameters: population of technical interns of 400,000, percentage of high occupational competence of 88%, and confidence limits of 3%, based on a previous study [24]. We estimated a minimum sample size of 445.

### 2.3. Recruitment Procedure

We recruited trainees through their supervising organizations and various social media platforms using a convenience sampling method. From the list of contacts published online, we communicated with supervising organizations and non-governmental organizations (NGOs). We sent the link to the survey platform to organizations that agreed to collaborate with us. The organizations shared the link via email or posted the QR code to inform the trainees about the survey.

### 2.4. Variables and Measurement

#### 2.4.1. Outcome: Mental Well-Being

We measured the mental well-being of trainees using the 5-item World Health Organization Well-Being Index (WHO-5). The WHO-5 is a short and generic global rating scale that measures subjective mental well-being. The raw score ranging from 0 to 25 is multiplied by 4 to give the final score from 0 representing the worst imaginable well-being to 100 representing the best imaginable well-being [25]. Technical intern trainees rated how well each of the five statements applied to them when considering the last 14 days. Higher scores indicate a higher level of mental well-being. The psychometric properties of validity and reliability are good to excellent in the Chinese version among adult patients in a pain clinic [26]. We conducted face validity, content validity, and reliability testing of the Vietnamese version of the WHO-5. The Cronbach’s *α* of the Vietnamese version of the WHO-5 for this study was 0.70.

#### 2.4.2. Exposure: Occupational Competence

Occupational competence was measured using Occupational Self-Assessment (OSA) version 2.2. The scale measures clients’ perceived competence in daily activities and the occupations they consider important based on 21 occupations [27]. The trainees first scored their level of competence for each of the 21 occupations on a four-point scale: “I have a lot of problems doing this (1)”, “I have some difficulty doing this (2)”, “I do this well (3)”, and “I do this extremely well (4)”. The total score of competence ranged from 21 to 84. We converted the raw scores of competence into interval measures using the OSA competence key forms, and the scores ranged from 0 to 100 [28]. A higher interval score indicates a higher level of perceived occupational competence.

In addition, OSA measures the importance of each occupation. These ratings indicate what the trainees valued in their daily lives. The trainees scored the importance of each occupation on a four-point scale: “this is not important to me (1)”, “this is important to me (2)”, “this is more important to me (3)”, and “this is most important to me (4)”. For importance, we analyzed the percentage of trainees who indicated an occupation as more important (3) or most important (4), as opposed to less important (2) or least important (1). The scores for importance were analyzed only descriptively, as our research purpose was to investigate the relationship between occupational competence and mental well-being. OSA has previously demonstrated good structural validity and internal consistency [29,30]. The Chinese version of the OSA has demonstrated good construct validity and moderate concurrent validity among patients with schizophrenia [31]. OSA has not yet been translated into Vietnamese. We performed pretesting to assess each version of the questions and correct them accordingly to improve content validity. We conducted face validity testing with bilingual experts and checked the reliability of the Vietnamese version of the OSA. The Cronbach’s *α* of the Vietnamese version of the OSA in this study was 0.88.

#### 2.4.3. Socio-Demographic Characteristics and Other Variables

Trainees responded to questions regarding socio-demographic factors, including gender, age, completed education, monthly income, living arrangement, length of stay, job type, prefecture of living, and daily working hours. For age, we asked participants for years of age. Educational qualification was categorized into three groups: middle school, high school, and university or above. Concerning monthly income, trainees responded with the actual amount that they received in the Japanese yen in the month prior to data collection. We categorized the living arrangement into three groups: living alone, living with family, and living with others (e.g., friends or coworkers). The length of stay was categorized into six groups: less than one year, one to less than two years, two to less than three years, three to less than four years, four to less than five years, and five years or more. Job type was categorized into five categories: farming and fisheries, manufacturing, construction, elderly care, and other services. The trainees chose their prefectures to live from a drop-down list. For working hours, trainees responded with their average work hours per day.

We also measured trainees’ Japanese language proficiency [32], self-rated health [33], and social support [34], in addition to socio-demographic characteristics, as these variables had been shown to have a relationship with mental well-being. We also included the question on the impact of COVID-19 on job security and income changes. We asked whether their income had changed before or after COVID-19, and whether their work had been continued, suspended, or terminated.

### 2.5. Data Collection

We created an online survey using Google Forms, consisting of 62 items with a landing page, consent form, and a screening question. We posted a link on social networking services (SNS) on the Internet, including group accounts of leaders of foreign support groups and organizations on Facebook and Twitter, with their permission. The trainees completed the survey on a smartphone, tablet, or personal computer. The trainees could review the study’s objectives, data collection procedure, analysis, storage, and privacy protection on the survey website’s landing page. We created a screening question to exclude those who did not meet the inclusion criteria. The survey was conducted from 11 August 2020 to 30 September 2020. In the case of a paper-based survey, collaborating organizations distributed the survey and stamped envelopes to trainees willing to participate. The trainees mailed the surveys to the researchers’ institutions. At the end of the web- or paper-based survey, the trainees could choose to receive an incentive of a 500 Japanese yen (equivalent to about 4.53 US dollars in July 2021) gift card by providing their names and mailing addresses.

### 2.6. Data Analysis

Descriptive statistics were used to summarize the trainees’ basic characteristics and identify their perceptions of occupational competence and their importance. We performed a hierarchical regression analysis to identify factors associated with trainees’ mental well-being. In Model 1, we included occupational competence only. In Model 2, we included socio-demographic characteristics, language proficiency, and occupational competence. In Model 3, we included social support, self-rated health, socio-demographic characteristics, language proficiency, and occupational competence. In Model 4 (full model), we included the effect of COVID-19 on job security and income, social support, self-rated health, socio-demographic characteristics, language proficiency, and occupational competence. No multicollinearity was observed in any of the models. The level of significance was set at *p* = 0.05. We used heteroskedasticity-robust standard error to address heteroskedasticity in the linear regression models. We used Stata.13.1 (StataCorp L.P., Collage Station, TX, USA) for data analysis.

## 3. Results

### 3.1. Characteristics of Participants

In total, we recruited 459 trainees through 28 supervising organizations and three NGOs. Of them, 356 and 103 trainees provided web-based and paper-based survey answers, respectively. Three trainees in the web-based entries declined to participate. After data cleaning, we found that 73 entries were missing substantial data. We excluded these entries from further analysis, leaving 383 valid responses (web = 286; paper = 97; Figure 1).

Table 1 shows the socio-demographic characteristics of the participants. Of the 383 trainees, 297 (77.5%) were Vietnamese, and 86 (22.5%) were Chinese. More women than men participated in the study (59.5% vs. 40.5%). The mean age was 24.0 years (standard deviation [SD] 6.2), and 70.2% had completed high school level education. Regarding working conditions, 70.5% of the trainees earned less than 150,000 yen (approx. 1440 US dollars) within a month, and 77.8% worked for eight hours a day. As for job categories, more than half of them worked in manufacturing (56.7%), followed by construction (32.9%), care work/services (7.3%), and farming/fishery (3.1%).

Regarding the length of stay in Japan, 60.6% of them had stayed in Japan for less than two years, and 92.4% lived with their family or others. Nearly half of them lived in the Kanto area (46.0%), followed by Kinki (20.6%) and Chubu (20.1%). Concerning their Japanese language proficiency, about half of them reported “none or little,” and the other half reported “moderate or fluent.” Regarding the level of social support, 31.9% of them received social support “usually” or “always,” while 20.1% “rarely” or “never” received it. Most reported their health as “good or very good” (96.6%). Their mean WHO-5 score was 67.5 (SD 21.3), and their mean OSA competence score was 58.1 (SD 8.5). Both scores ranged from 0 to 100. (Appendix A: Table A1 for WHO-5).

### 3.2. Occupational Competence and Importance

Figure 2 shows the percentages of trainees who perceive a higher level of difficulty and importance in each OSA occupation item. Regarding occupational competence, 30.6% felt difficulty expressing themselves to others, and 27.4% experienced difficulty in accomplishing what they set out to do. A relatively small percentage of them had difficulties in taking care of themselves (4.4%) and taking care of the place they lived in (5.2%). Regarding importance, more than half of them rated taking care of themselves (54.3%) as important. Other highly valued occupation items were working towards their goals (50.4%) and managing their finances (49.9%). (Appendix B: Figure A1 for correlations between competence and importance of occupation items).

### 3.3. Factors Associated with Mental Well-Being of Technical Intern Trainees in Japan

Table 2 presents the results of the hierarchical regression analyses. According to the final multiple regression model, a higher level of occupational competence (B = 0.76; 95% CI = 0.52, 1.00) was associated with a higher level of mental well-being. Hierarchical regression analyses revealed that competence in daily activities remained significant predictors of mental well-being after controlling for the effect of other variables. The standardized coefficient of OSA competence in Model 4 was 0.31. It shows that mental well-being is higher by 0.31 standard deviations on average, controlling for other independent variables, when OSA competence is higher by one standard deviation. The size of its beta coefficient was one of the largest among the independent variables in the final model.

The following factors were also associated with higher levels of mental well-being: being Vietnamese (B = 17.75; 95% CI = 12.14, 23.36), of older age (B = 0.37; 95% CI = 0.16, 0.59), and having good or very good health (B = 27.24; 95% CI = 10.97, 43.52). In contrast, the trainees who received social support “sometimes” (B = −7.43; 95% CI = −11.48, −3.39) and “never/rarely” (B = −16.81; 95% CI = −26.43, −7.19) were associated with a lower level of mental well-being than those who always or usually received social support. When we analyzed the interaction terms, nationality did not affect the relationship between mental well-being and other variables. (Appendix C: Table A2 for interaction term analysis) When incorporated into multiple linear regression analysis, interactions were not significant nor did they contribute to the model’s better fit.

## 4. Discussion

In this study, the trainees found it difficult to express themselves to others and accomplish what they set out to do while they were in Japan. They highly valued taking care of themselves, working towards their goals, and managing their finances. Higher levels of competence in daily activities were associated with higher levels of mental well-being among trainees. Other factors associated with higher levels of mental well-being were nationality, older age, and satisfactory health. However, receiving a lower level of social support was associated with a lower level of mental well-being, compared to receiving a higher level of social support.

Our results showed that the trainees experienced difficulty expressing themselves in their daily lives in Japan. Although they attended Japanese language courses [35], they did not pay attention to self-expression in daily life. Self-expression is important because it has protective effects on mental well-being [36]. The results imply a need for opportunities for self-expression using a variety of activities, rather than just aiming to improve language skills. Using creative activities to express oneself may help in building social networks in conjunction with improving mental well-being.

In this study, the trainees highly valued taking care of themselves, managing their finances, and working towards their goals. Since trainees’ primary purpose of coming to Japan was to acquire skills and earn money for their families [37], it is crucial for them to take care of themselves in order to work fully, earn money, and achieve their goals until their employment term ends. Almost half of the trainees perceived accomplishing their goals as important, but almost one-fourth perceived this to be difficult to perform. This large gap needs to be emphasized [28] and explored further because it indicates trainees’ greatest dissatisfaction with the occupational item. Attaining intrinsic goals, such as personal growth [38,39], is positively related to acculturation and life satisfaction and negatively related to anxiety and depression among the migrant population [40]. Supporting trainees to identify and accomplish their goals might support their acculturation and mental well-being.

Trainees competent in their daily activities were more likely to experience a higher level of mental well-being. Even though trainees tend to be exposed to migration-related health hazards and restrictions [5,7] under the Technical Intern Training Program, their competence in daily activities can have a protective role in their mental well-being. This finding accords with previous literature on non-migrant populations. For example, Israeli mothers who were competent in their daily activities reported higher levels of mental well-being [13]. Even though the trainees were exposed to new cultures and lifestyles, different from their respective countries, they acquired a higher level of competence in daily activities when they had opportunities to participate in various daily activities [10] in Japan. Consequently, this repeated participation in a wide array of daily activities could lead to a higher level of mental well-being in this population.

Being Vietnamese was associated with a higher level of mental well-being than being Chinese. The results of this study could not conclude any differences among nationalities, but some similar findings have been reported. For example, in Australia, Vietnamese migrant workers were less likely to report being bullied in the workplace than Chinese. On the other hand, Chinese migrant workers showed an increased likelihood of low job security compared to their counterparts who worked the same hours [2]. Moreover, in Japan, 40.6% of Chinese workers reported suffering from depression [41]. In this study, nationality did not affect the relationship between mental well-being and other variables based on the interaction terms analysis. Chinese migrant workers might have different factors that make them more likely to feel stressed in a different culture, but we did not identify them in this study.

Older trainees showed higher levels of mental well-being. A similar finding was reported among Mexican immigrants (aged 18–25) in the United States, which demonstrated that there is a lower risk of depression or anxiety among older immigrants than among younger immigrants [42]. In Australia [43] and China [44], older migrants showed higher levels of mental well-being. Several factors might have contributed to the higher level of mental well-being. First, older trainees had a higher level of competence in daily activities because they could have learned different skills in their home countries before they migrated to Japan as trainees. Second, the discrepancy between pre-migratory expectations and post-migratory reality could lead to a sense of loss of purpose and a negative impact on mental well-being among young trainees [45]. The trainees’ length of stay was not associated with mental well-being after adjusting for other variables in the regression model. Therefore, being older was one of the key factors of staying in Japan, with better mental well-being among the older trainees regardless of their length of stay in Japan.

The trainees showed a higher level of mental well-being when they perceived themselves as having good health. This finding is consistent with other studies, as researchers have found a strong link between physical and mental health [46]. For instance, Kurdish migrants in Sweden, with poor self-rated health, had more than three times higher odds of having low mental well-being [47]. The trainees in good health were more likely to feel content because they could work fully to accomplish their job requirements and master skills. Being healthy may bring a sense of hope that one can have life goals for the future.

Trainees who perceived a lower level of social support reported a lower level of mental well-being. In systematic reviews, a higher level of social support was associated with better mental well-being among migrant populations in Japan [48] and worldwide [49]. In addition, in the United States, a level of social support was associated with Latino [50] and Asian-American migrant workers’ [51] levels of mental well-being. Social support reduces the impact of acculturative stress [52], and migrants receiving social support are more likely to access health care services [53]. Therefore, trainees with a higher level of social support might feel less stressed and could access mental health care early, resulting in better mental well-being. The trainees who felt that they were receiving adequate social support could have trustful relationships with others and participate in a variety of daily activities with a sense of security in their communities.

This study has several limitations. First, the Vietnamese versions of OSA and WHO-5 were translated within our research team, whereas the Chinese versions were validated in previous studies. We conducted pretesting on Vietnamese trainees to measure the reliability of the scale and performed face-validity testing with bilingual experts. Although the internal consistency was acceptable for WHO-5 (*α* = 0.70) and good for OSA (*α* = 0.88), the differences in cultural backgrounds or language translations might have led to bias in different interpretations of the questionnaire. Second, missing data and non-response could cause information bias because trainees with more missing data might have different occupational competences and mental well-being. Third, we used a convenience sampling method, so the obtained sample might not represent the entire trainee population. However, we tried to overcome this limitation by reaching trainees in 33 prefectures, which covered 70% of the prefectures in Japan. We also used a paper-based survey to include trainees who did not have access to online social networking sites. Fourth, this study has a relatively small sample size. That may limit statistical power. Nevertheless, the association between OSA competence and WHO-5 was robust in terms of including additional independent variables in this study. Finally, there might be other factors that were not covered in the study, such as community engagement [54], pre-migratory mental health status [43], and culture competence [55], which might also affect their mental well-being. These factors must be considered in future research as they may play a significant role in the well-being of migrants.

Despite these limitations, this study has several strengths. This study was the first to assess occupational competence and importance. Moreover, it was the first to examine the association between occupational competence and mental well-being among technical intern trainees in Japan. It also brought new attention to the competence and importance of daily activities among migrant populations. The findings of this study may be utilized to create a supportive environment for trainees in Japan.

The findings of this study have important implications for mental health promotion policies to focus more on competence in daily activities among trainees in Japan. Community programs will need to provide a supportive and enabling environment for trainees to increase their competence. For example, community program providers may incorporate more opportunities for trainees to express themselves and identify their goals in their daily lives.

## 5. Conclusions

Trainees experienced difficulty in expressing themselves and accomplishing their goals. They highly valued taking care of themselves, working towards their goals, and managing their finances. The trainees who had higher competence in daily activities showed a higher level of mental well-being. They might benefit from programs that focus more on investigating personal competence in daily activities to ensure sufficient mental well-being regardless of nationality, age, self-rated health, and social support.

## Figures and Tables

**Figure 1 ijerph-19-03189-f001:**
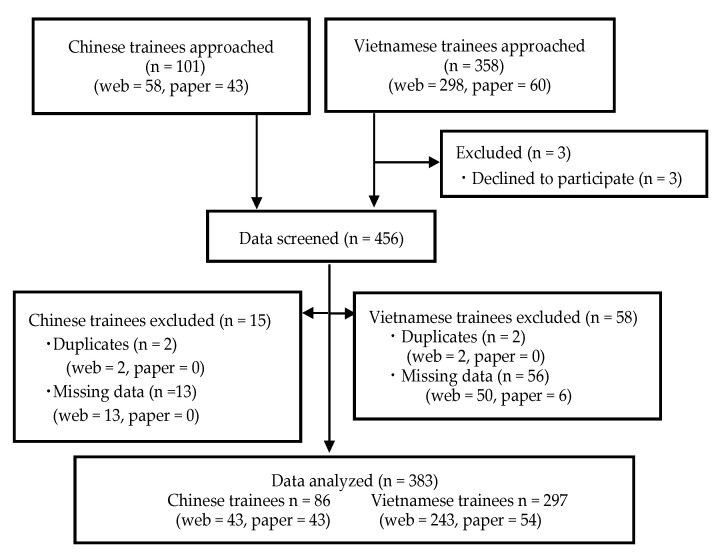
Flow diagram of study participants.

**Figure 2 ijerph-19-03189-f002:**
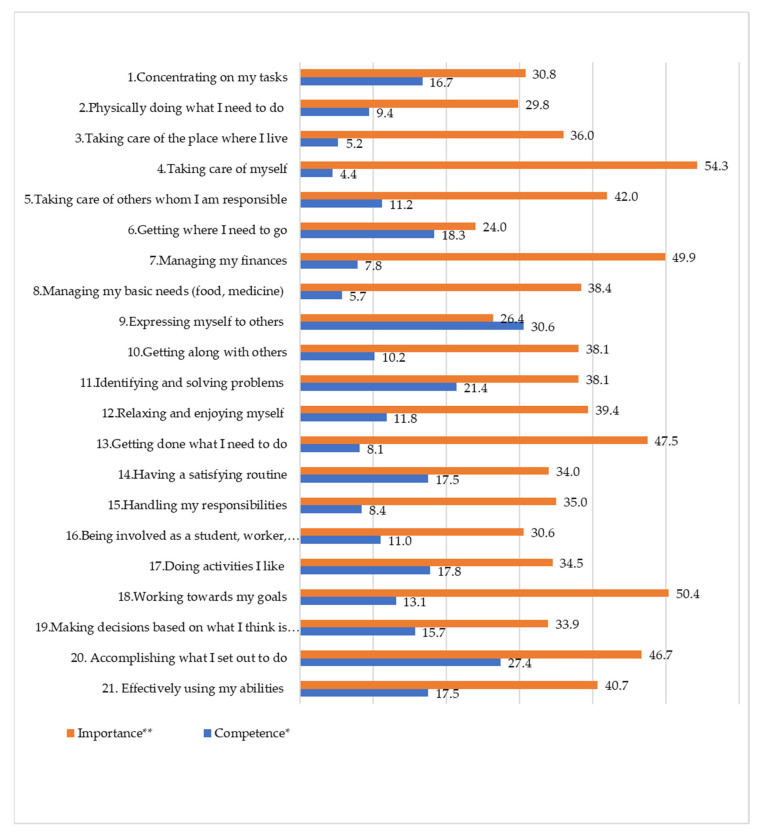
Proportions of competence and importance of occupation items. * A proportion of trainees who answered “I have a lot of problems doing this” or “I have some difficulty doing this” in a 4-point Likert scale question. ** A proportion of trainee who answered “this is more important to me” or “this is most important to me” in a 4-point Likert scale question.

**Table 1 ijerph-19-03189-t001:** Characteristics of technical intern trainees.

	Total (*n* = 383)		Chinese (*n* = 86)		Vietnamese (*n* = 297)	
	*n*	%	*n*	%	*n*	%
Age, mean (SD)	24.0 (6.2)		30.5 (7.6)		22.2 (4.3)	
Gender						
Man	155	40.5	48	55.8	107	36.0
Woman	228	59.5	38	44.2	190	64.0
Completed education						
Middle school	32	8.4	28	32.6	4	1.4
High school	269	70.2	46	53.5	223	75.1
University or above	82	21.4	12	14.0	70	23.6
Monthly income (Yen)						
Less than 150,000	270	70.5	31	36.0	239	80.5
150,000 and more	113	29.5	55	64.0	58	19.5
Living situation						
Live alone	29	7.6	8	9.3	21	7.1
Live with family or others	354	92.4	78	90.7	276	92.9
Length of Stay						
Less than 1 year	114	29.8	17	19.8	97	32.7
1–2 years	118	30.8	38	44.2	80	26.9
2–3 years	94	24.5	17	19.8	77	25.9
More than 3 years	57	14.9	14	16.3	43	14.5
Job						
Farming/fishery	12	3.1	3	3.5	9	3.0
Manufacturing	217	56.7	19	22.1	198	66.7
Construction	126	32.9	47	54.7	79	26.6
Care worker/Services	28	7.3	17	19.8	11	3.7
Prefecture						
Hokkaido/Tohoku	27	7.1	11	12.8	16	5.4
Kanto	176	46.0	40	46.5	136	45.8
Chubu	77	20.1	10	11.6	67	22.6
Kinki	79	20.6	17	19.7	62	20.9
Others	24	6.3	8	9.3	16	5.4
Hours of work (hours/day)						
Less than 8 h	24	6.3	5	5.8	19	6.40
8 h	298	77.8	67	77.9	231	77.8
More than 8 h	61	15.9	14	16.3	47	15.8
Language level						
None/A little	192	50.1	69	80.2	123	41.4
Moderate/Fluent	191	49.9	17	19.8	174	58.6
Social support						
Never/Rarely	77	20.1	16	18.6	61	20.5
Sometimes	184	48.0	37	43.0	147	49.5
Always/Usually	122	31.9	33	38.4	89	30.0
Self-rated health						
Very bad/Bad	13	3.4	7	8.1	6	2.0
Very good/Good	370	96.6	79	91.9	291	98.0
COVID-19 impact on:						
Job security						
Sustained	348	90.9	83	96.5	265	89.2
Suspended/Terminated	35	9.0	3	3.5	32	10.8
Income change						
Increase	66	17.2	16	18.6	50	16.8
Decrease	107	27.9	33	38.4	74	24.9
Same	210	54.8	37	43.0	173	58.3
WHO-5, mean (SD)	67.5 (21.3)		57.5 (23.4)		70.4 (19.8)	
OSA competence, mean (SD)	58.1 (8.5)		59.3 (9.4)		57.8 (8.2)	

**Table 2 ijerph-19-03189-t002:** Hierarchical regression analysis predicting the mental well-being among technical intern trainees in Japan (*n* = 383).

Variables	Model 1	Model 2	Model 3	Model 4 (Final Model)		
	Coef.	Coef.	Coef.	Coef.	(95% CI)	β
OSA competence	0.93 ***	0.99 ***	0.76 **	0.76 **	(0.52, 1.00)	0.31
Nationality (ref. Chinese)						
Vietnamese		19.35 ***	17.33 **	17.75 **	(12.14, 23.36)	0.35
Age		0.44 **	0.38 *	0.37 *	(0.16, 0.59)	0.11
Gender (ref. Woman)						
Man		−2.70	−1.50	−1.51	(−3.78, 0.76)	−0.03
Completed education						
(ref. Middle school)						
High school		−6.81	−5.60	−4.81	(−10.91, 1.29)	−0.10
University or above		−1.45	−2.07	−1.18	(−5.19, 2.83)	−0.02
Monthly income (Yen)						
(ref. Less than 149,999)						
150,000 and more		0.93	0.95	1.03	(−4.39, 6.44)	0.02
Living arrangement (ref. Live alone)						
Live with family/others		4.24	4.85	4.51	(−6.01, 15.02)	0.06
Length of stay (ref. More than 3 years)						
Less than 1 year		0.91	2.55	3.11	(−1.41, 7.63)	0.07
1–2 years		0.26	0.75	1.00	(−0.36, 2.36)	0.02
2–3 years		4.60	6.11	6.06	(−0.81, 12.98)	0.12
Language level (ref. None/A little)						
Moderate/Fluent		−0.55	−0.41	−0.35	(−5.46, 4.77)	−0.01
Social support (ref. Always/Usually)						
Sometimes			−7.26 **	−7.43 **	(−11.48, −3.39)	−0.17
Never/Rarely			−16.41 **	−16.81 **	(−26.43, −7.19)	−0.32
Self-rated health (ref. Very bad/Bad)						
Very good/Good			27.35 *	27.24 *	(10.97, 43.52)	0.23
COVID-19 impact on:						
Job security						
(ref. Suspended/Terminated)						
Sustained				4.22	(−9.79, 18.23)	0.06
Income change (ref. Decreased)						
Increase				−4.17	(−13.43, 5.10)	−0.07
Same				−3.05	(−8.67, 2.57)	−0.07
R^2^	0.14	0.26	0.39	0.40		
ΔR^2^		0.12	0.13	0.01		

Coef.—Unstandardized regression coefficient (B); CI—Confidence interval; β—Standardized beta coefficient; COVID-19—Coronavirus disease 2019; OSA—Occupational Self-Assessment; R^2^—R-squared: ΔR^2^—Incremental increase in R^2^; Statistical significance indicated by * *p* < 0.05; ** *p* < 0.01; *** *p* < 0.001.

## Data Availability

Anonymized data may be made available upon request. Requests should be directed to Akira Shibanuma (shibanuma@m.u-tokyo.ac.jp).

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
