# Peer review of "Competence in Daily Activities and Mental Well-Being among Technical Intern Trainees in Japan: A Cross-Sectional Study"

_ijerph, 2022, doi:10.3390/ijerph19063189_

Round 1
Reviewer 1 Report
Review
This paper examines the relationship between immigrant workers' mental well-being and OSA competence. It is evaluated as a timely research in these days of frequent international movement of labor. Nevertheless, the following needs to be improved.
- There are too many independent variables compared to the observations of data. For this reason, there is a possibility that statistically significant results can be derived despite the small difference. For example, in the case of Chinese sample, the total number is 86, but there are 4 cells with a size of 5 or less. Even in the case of Vietnam sample, there are a cell with 5 or less
- For me, it is not understood whether the mean value of the World Health Organization Well-Being Index (WHO-5) used as the dependent variable in Table 1 is 67.5. The maximum value of the 5 items is 25, so the WHO-5 value has a value between 0-25. There is no explanation for this.
Please show the normality test result of the dependent variable (mental well-being score).
- Figure 2 does not seem to be clearer than Appendix B. In Figure 2, what are the authors trying to explain.
- In Table 2 showing the results of regression analysis, why is OSA competence not included in all models but only in model 4? This is especially true because the purpose of this paper is to report the relationship between mental well-being and OSA competence. It is necessary to evaluate how the relationship between mental well-being and OSA competence changes according to the difference in independent variables and how robust it is.
- As mentioned above, since many independent variables are used with a small sample, have you not considered the possibility that a statistically significant result can be derived even with a very small value change? Revises such as reconsider the number of independent variables or readjust the categories of independent variables seem necessary.
- The interpretation of the coefficients of working hours and income in discussion section (line of 289-296) is not clear. It seems to need a more comprehensive interpretation and analysis such as the interaction effect
- The description of 289-296 in the discussion section needs to be revised. What is not statistically significant means that there is no difference between them. And it seems unreasonable to use this as an example with a difference. Also, comparison examples between Chinese and Japanese are not appropriate.
- Despite of a statistically significant difference between mental well-being and OSA competence, the regression coefficient did not appear to be large compared to other variables. Considering the average value and SD of OSA competence in Table 1, please describe the meaning of this regression coefficient value quantitatively
Reviewer 2 Report
The authors of the article raise the problem of the relationship between mental health and competence of migrants in Japan. It is quite obvious that the base of the study can be any country, and the specifics of the host country can only be associated with the use of special adaptation measures for migrants and the general social conditions of the host country (for example, cordiality or alertness). However, socio-psychological factors are the most significant for the adaptation of migrants. However, the article does not take into account cultural variables, such as values ​​or cultural dimensions. Relations with people in the country of origin, in the host country, with the diaspora and with the local population are important. The authors focused on self-assessment of everyday life competencies. It seems obvious that people who consider themselves more competent in everyday life will also be more prosperous. Meanwhile, the effects of employment and support, etc., should have been more clearly spelled out, it would have been advisable to place figures with these effects in the article, and also discuss the very high level of discrepancy between the significance and self-assessment of competencies.
Round 2
Reviewer 1 Report
Despite of a statistically significant difference between mental well-being and OSA competence, the regression coefficient did not appear to be large compared to other variables. Considering the average value and SD of OSA competence in Table 1, please describe the meaning of this regression coefficient value qualitatively.
The standardized coefficient of OSA competence in Model 4 was 0.76 It shows that WHO-5 is higher by 0.76 standard deviations on average, controlling for other independent variables, when OSA competence is higher by one standard deviation.
à This is not the response I expected. The regression coefficient does not appear to be large compared to other variables of coefficients, such as education and social support. Being statistically significant and having an effect on dependent variable have different meanings. Please describe this difference qualitatively.
